# Anomeric memory of the glycosidic bond upon fragmentation and its consequences for carbohydrate sequencing

Baptiste Schindler [1,2,3], Loïc Barnes[1,2,3], Gina Renois[1,2,3], Christopher Gray[4], Stéphane Chambert[1,2,5], Sébastien Fort[6,7], Sabine Flitsch[4], Claire Loison[1,2,3], Abdul-Rahman Allouche [1,2,3] & Isabelle Compagnon[1,2,3,8]

Deciphering the carbohydrate alphabet is problematic due to its unique complexity among biomolecules. Strikingly, routine sequencing technologies—which are available for proteins and DNA and have revolutionised biology—do not exist for carbohydrates. This lack of structural tools is identified as a crucial bottleneck, limiting the full development of glycosciences and their considerable potential impact for the society. In this context, establishing generic carbohydrate sequencing methods is both a major scientific challenge and a strategic priority. Here we show that a hybrid analytical approach integrating molecular spectroscopy with mass spectrometry provides an adequate metric to resolve carbohydrate isomerisms, i.e the monosaccharide content, anomeric configuration, regiochemistry and stereochemistry of the glycosidic linkage. On the basis of the spectroscopic discrimination of MS fragments, we report the unexpected demonstration of the anomeric memory of the glycosidic bond upon fragmentation. This remarkable property is applied to de novo sequencing of underivatized oligosaccharides.

[1] Université de Lyon, F-69622 Lyon, France. [2] Université Lyon 1, Villeurbanne, France. [3] Institut Lumière Matière, UMR5306 Université Lyon 1-CNRS, Université de Lyon, 69622 Villeurbanne Cedex, France. [4] School of Chemistry & Manchester Institute of Biotechnology, The University of Manchester, 131 Princess Street, Manchester M1 7DN, UK. [5] Laboratoire de Chimie Organique et Bioorganique, INSA Lyon, CNRS, UMR5246, ICBMS, Bât. J. Verne, 20 Avenue A. Einstein, 69621 Villeurbanne Cedex, France. [6] Université de Grenoble Alpes, CERMAV, F-38000 Grenoble, France. [7] CNRS, CERMAV, F-38000 Grenoble, France. [8] Institut Universitaire de France IUF, 103 Blvd St Michel, 75005 Paris, France. Correspondence and requests for materials should be addressed to I.C. (email: isabelle.compagnon@univ-lyon1.fr)

The first principles of protein and DNA sequencing, which have been established between 1952 and 1977, have revolutionised modern biology and have even become amazingly popular among the non-scientific public. Yet, it is striking that routine, high-throughput sequencing technology is not available for oligosaccharides. This lack of carbohydrate-oriented structural tools, which is identified as a critical impairment to the full development of glycosciences[1, 2], is essentially due to their unique molecular complexity among natural biopolymers.

The basic concept of tandem-mass spectrometry (MS) analysis of a (bio-)polymer sequence rests on two foundational hypothesis: the molecular structure of fragmentation products must be translatable to the parent polymer through retention of structural features of the precursor; and one must have available an effective metric offering a sufficient level of structural detail of fragments to retrieve the structure of the precursor. These pre-requisites are verified for proteins and nucleic acids, which are linear biopolymers formed from a relatively small pool of often non-isomeric monomers, allowing the parent sequence to be deduced from $m/z$ alone[3]. The case of carbohydrates is more complex as both the monomer composition (most of which are epimers of one another), and the regiochemistry and stereochemistry of the inter-glycosidic linkages must be identified. Moreover, these isomeric carbohydrate structural features are not readily resolved using traditional MS techniques[4]. To date, the preservation of these carbohydrate structural features upon fragmentation is poorly understood. Domon and Costello[5] proposed a nomenclature for MS carbohydrate fragments, but their exact molecular structure was not verified, largely due to the lack of a universal metric with sufficient structural resolution to disambiguate all carbohydrate isomerisms.

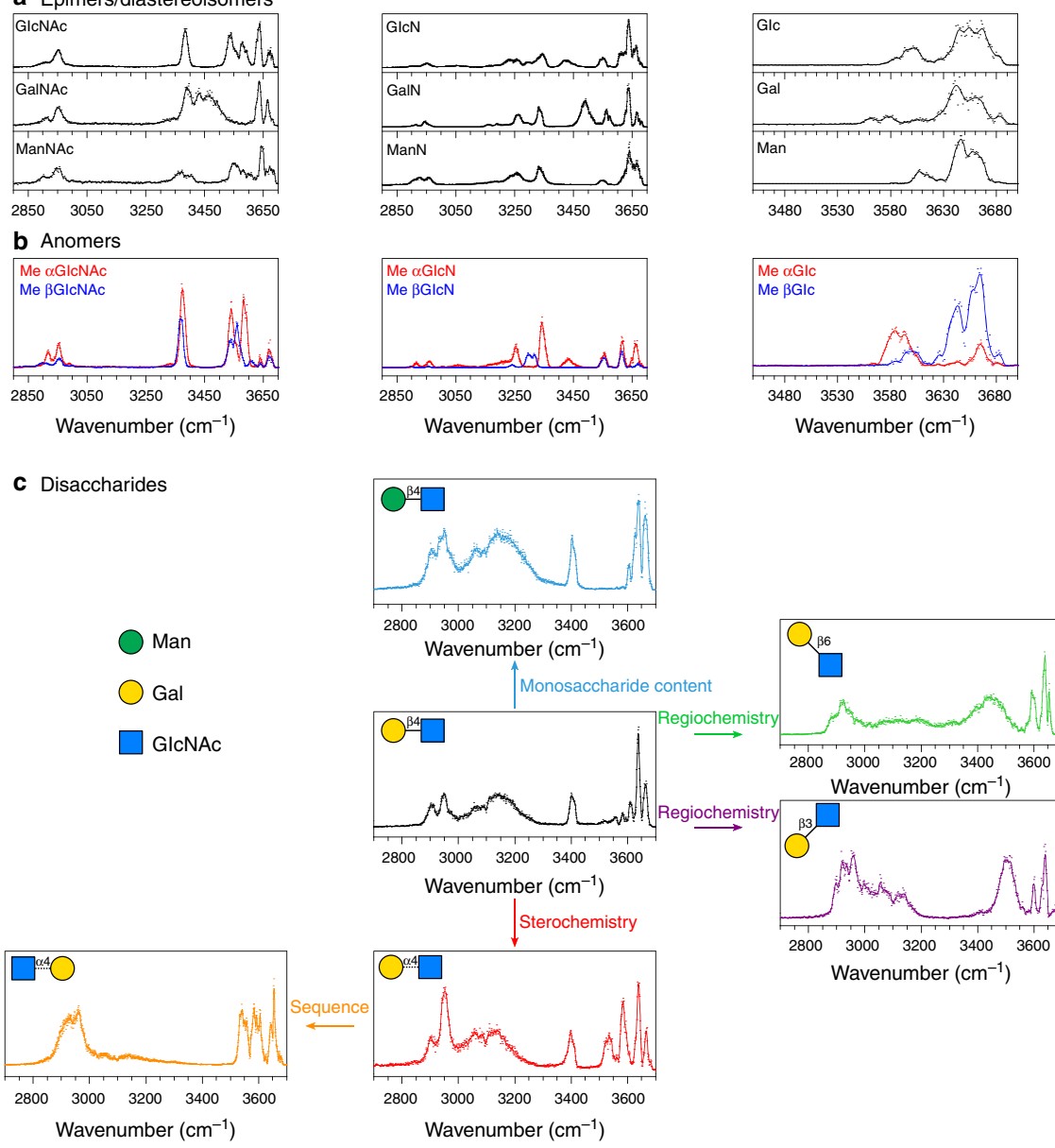

**Fig. 1** IRMPD signatures of monosaccharide and disaccharide standards. **a** *N*-acetylhexosamines, hexosamines and hexoses. **b** Individual methyl-blocked anomers of *N*-acetylglucosamine, glucosamine and glucose. **c** Isomers of GalGlcNAc: Galβ1,4GlcNAc (black); Manβ1,4GlcNAc (blue); Galβ1,3GlcNAc (purple); Galβ1,6GlcNAc (green); Galα1,4GlcNAc (red); GlcNAcα1,4 Gal (yellow). Symbols from ref. [40]

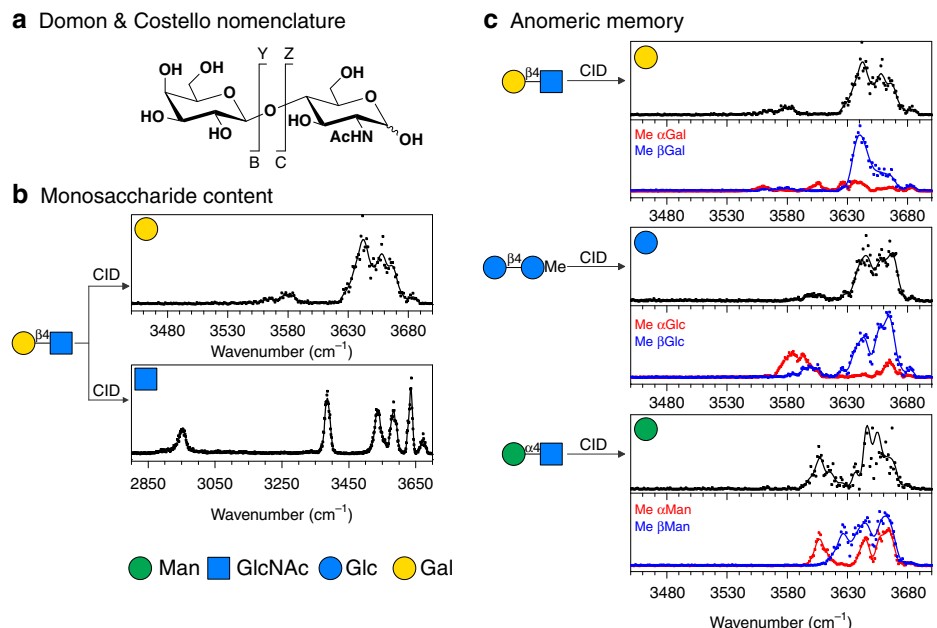

**Fig. 2** Spectroscopic analysis of disaccharides fragments. **a** Nomenclature of carbohydrates MS fragments, after Domon and Costello[5]. **b** Monosaccharide fragments of Galβ1,4GlcNAc. **c** Non-reducing fragments of Galβ1,4GlcNAc; Glcβ1,4GlcMe and Manα1,4GlcNAc. Methyl-anomers of Gal, Glc and Man are shown for comparison. Symbols from ref. [40]

Owing to these limitations, structural elucidation of unknown oligosaccharides from biological sources largely relies on NMR, which may imply months of extraction and purification of a sufficient amount of sample, with possible alteration of labile functional modifications. Alternatively, mass spectrometric analysis applies to natural samples but does not readily resolve all carbohydrate isomerisms. Whereas MS/MS analysis of cross-ring fragments is useful to resolve regiochemistry, the stereochemistry and the monosaccharide content remain ambiguous by MS analysis alone. This has stimulated considerable efforts to explore integrated hyphenated approaches. Such approaches aim at providing direct structural information on mass-selected ions without the recourse to chemical derivatisation of the analyte, or to a high level of chemical purification. In particular, ion mobility spectrometry (IMS) and field asymmetric ion mobility spectrometry (FAIMS) separation of isomeric carbohydrates were explored since 1997 using home-built FAIMS-MS or IMS-MS instruments[6–11]. More recently, the formidable potential of commercial IMS-MS instruments has been demonstrated[12–16]. Alternatively, Bendiak and colleagues[17] proposed an approach based on IRMPD spectroscopy (infrared multiple photon dissociation) to explore saccharides isomers in the free-electron laser spectral range (5–10 μm); whereas Pohl and colleagues[18, 19] have reported monosaccharide identification using kinetic measurements. So far however, none of these methods could resolve simultaneously all carbohydrates isomerisms.

Although the question of the universal metric for carbohydrates remains open, MS sequencing also requires that the structural properties of an oligomer are preserved upon fragmentation and can be retrieved via the structural analysis of its fragments. The Domon and Costello nomenclature for carbohydrate MS fragments suggests that monosaccharides fragments of type C and Y are intact monosaccharide units; and C fragments bear the anomeric configuration of the glycosidic bond of the precursor. This was however never formally demonstrated and these hypotheses need to be verified for the purpose of carbohydrate sequencing.

In this context, few groups have reported ion mobility exploration of carbohydrates fragments. Pagel in particular reported the identification of structurally informative patterns in glycans via the analysis of trisaccharide fragments[20, 21]. Others reported collision cross sections of monosaccharide fragments, but the results were difficult to relate to the collision cross sections of the corresponding monosaccharide standards[13, 22, 23]. Recently, we have reported that B and C fragments of glucose containing disaccharides exhibit precursor-dependant profiles[24]. So far, reliable discrimination of monosaccharide fragments and the identification of their anomeric configuration are limited by structural resolution of ion mobility spectrometry. As a direct consequence, the fundamental hypothesis underlying top–down carbohydrate analysis, that is the anomeric memory of C fragments suggested by Domon and Costello, remains to be investigated.

Here, we present the spectroscopic fingerprint in the 3 μm spectral range as a versatile carbohydrate metric. Its refined structural resolution allows simultaneous resolution of monosaccharide isomers, regio-isomers and stereo-isomers in an integrated instrument combining MS and IR spectroscopy. Using the resolving power of this metric, we explore the structure of carbohydrate fragments and we report the validated observation of anomeric memory of the glycosidic bond within C-fragments. Finally, we illustrate the potential of this remarkable property for de novo carbohydrate sequencing, with the full characterisation (i.e. degree of polymerisation, degree of acetylation and pattern of acetylation) of a crude sample of chito-oligosaccharides.

## Results

**Carbohydrate isomerisms resolution using IRMPD fingerprint.** In traditional IR spectroscopy, the 3 μm spectral range is of little molecular specificity due to the interferences with aqueous solvents. For carbohydrates in the gas phase however, this spectral range is highly specific, as established by the pioneering work of Simons[25] on jet-cooled, synthetically phenyl-grafted

monosaccharides and oligosaccharides[26, 27]. With the integration of IR spectroscopy to mass spectrometry (IRMPD spectroscopy), the applicability of this highly relevant spectral range has been generalised to non-derivatised carbohydrates. Several groups have reported diagnostic fingerprints of monosaccharides isomers in this spectral range, including hexoses[28, 29], hexuronic acids[30, 31], N-acetyhexosamines[32, 33], and sulfated and phosphorylated monosaccharides[34–36]. Recently, we have also reported the spectroscopic identification of oligosaccharide isomers[36, 37].

Here we further generalise the 3 μm fingerprint as an adequate metric to resolve carbohydrate isomerisms by briefly reviewing the gas phase IR signatures of a range of monosaccharide and disaccharide standards.

As shown in Fig. 1a, epimers and/or diastereoisomers of the most abundant natural hexoses, N-acetylhexosamines and hexosamines can unambiguously be discriminated by their spectroscopic signatures. It is possible to further assign the spectroscopic features to individual hydroxyl and NH groups and to carry out complete conformational analysis by comparison with quantum chemistry simulations (Supplementary Fig. 1). Figure 1b shows the reference spectra of pairs of anomers (blocked by 1-O-methylation), which display distinctive signatures. Interestingly, these data can be used to identify the contributions of both anomers to the spectrum of a natural monosaccharide (Supplementary Fig. 2). The IR fingerprint is also diagnostic for the stereochemistry of the glycosidic bond, as shown in Fig. 1c for Galβ1,4GlcNAc (black trace) and Galα1,4GlcNAc (red trace) and for an inversion of sequence

(GlcNAcα1,4Gal, yellow trace). A change of monosaccharide content—known to be undetectable by ion mobility spectrometry[12]—results in a small, yet observable change of the IR fingerprint around 3600 cm⁻¹ (blue trace). Finally, the regioisomers of β-linked GalGlcNAc have well-resolved IR signatures (purple, black and green traces), in contrast with recent ion mobility data reported by Pagel and colleagues[21] Thus, IRMPD spectroscopy offers an efficient alternative to more elaborate spectroscopic schemes—such as this initially proposed by John Simons, or the cryogenic schemes recently proposed by Rizzo and colleagues[38] and Pagel and colleagues[39] for the distinction of carbohydrate isomers.

These results highlight the complementarity of IR spectroscopy with ion mobility spectrometry and qualify the IR fingerprint in the 3 μm spectral range as a powerful metric for structural characterisation of carbohydrates. Nevertheless, two limitations of this approach for glycoanalysis can be anticipated. First, the identification of an oligosaccharide will rely on the availability of the corresponding standard. Second, the structural resolution of the MS/IR fingerprint tends to decrease as the size of the oligosaccharide increases[37]. To overcome these two limitations, it is essential to adopt a top–down sequencing approach, as described in the next section.

**Spectroscopic analysis of MS fragmentation products**. IR spectroscopy integrated to MS offers the possibility to perform spectroscopic analysis building block by building block after

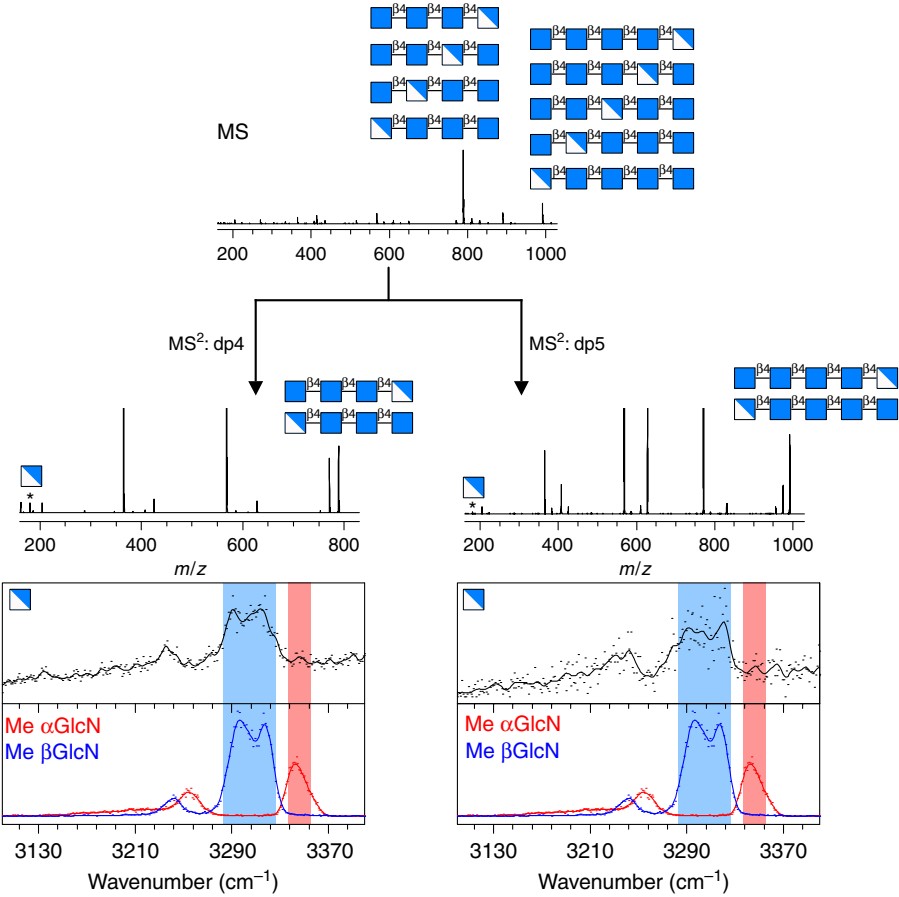

**Fig. 3** Illustration of the carbohydrate-sequencing procedure in the case of a crude mixture of chito-oligosaccharides: the degree of polymerisation and the fraction of acetylation are resolved by MS; the set of candidate sequences is reduced by MS/MS analysis; the sequence is fully resolved by spectroscopic analysis of the GlcN fragment. Symbols from ref. [40]

fragmentation by MS/MS. To verify that sequence information is conserved within the MS fragments proposed by Domon and Costello (Fig. 2a), we assessed their structure by comparison of their spectroscopic signatures with the reference spectra of monosaccharides standards.

**Elucidation of the monosaccharide content**. MS/MS analysis of Galβ1,4GlcNAc shows two monosaccharide moieties: a hexose (C fragment) and an *N*-acetylhexosamine (Y fragment), but does not readily indicate their nature. To resolve the monosaccharide content, the IR fingerprints of these two MS fragments are measured, as shown in Fig. 2b, and compared with the reference IR signatures of hexose and *N*-acetylhexosamine standards shown in Fig. 1a. The fingerprints of the C and Y fragments unambiguously match those of galactose and *N*-acetylglucosamine, respectively. This validates that the monosaccharide content of an oligosaccharide can be retrieved by IR analysis of its fragments. Remarkably, identification only relies on the comparison with a reduced database of monosaccharide standards and does not require a—potentially much larger—set of disaccharide references.

**Anomeric memory of the glycosidic bond upon fragmentation**. A closer examination of the galactose C fragment reveals that its fingerprint is not exactly identical to that of the galactose standard shown in Fig. 1a. Indeed, the small band at 3605 cm$^{-1}$ present in the spectrum of the standard is absent in the spectrum of the fragment. Further comparison with the reference fingerprints of the individual methyl-blocked anomers αGal and βGal is shown in Fig. 2c. It appears that this feature at 3605 cm$^{-1}$ is diagnostic of the α anomer, and its absence in the spectrum of the C fragment indicates that the galactose moiety retains a pure β character after fragmentation of the glycosidic bond. Similarly, Fig. 2c shows that the glucose moiety of Glcβ1,4GlcMe retains a pure β character (by comparison with methyl-blocked αGlc and βGlc standards); and the mannose moiety of Manα1,4GlcNAc retains a pure α character (by comparison with methyl-blocked αMan and βMan standards).

This observation constitutes evidence that the anomeric configuration of monosaccharide fragments holds the memory of the stereochemistry of the glycosidic bond. The reducing monosaccharide is the exception: it displays both α and β characters, as seen in the previous section. As a consequence of this important property, both the monosaccharide content and the stereochemistry of the glycosidic bond can be resolved by spectroscopic analysis of the monosaccharide fragments, which is an essential finding in the prospective of carbohydrate sequencing.

**Application to carbohydrate sequencing**. By extension, the monosaccharide content and the stereochemistry of an oligosaccharide can be elucidated by spectroscopic analysis of its monosaccharide fragments. To that end, IR spectroscopy is performed in a mass spectrometer equipped with an ion trap with MS$^n$ capability. In this configuration monosaccharide fragments are produced sequentially from the end and the order of the sequence is preserved in the process[41]. Thus, step-by-step spectroscopic analysis of the monosaccharide content and stereochemistry can be used to resolve the carbohydrate sequence.

As an illustration, we apply this sequencing approach to the resolution of chito-oligosaccharide sequences in a crude sample. This class of linear oligosaccharides consists of β1,4-linked GlcNAc and GlcN units. A comprehensive description of their structure includes their degree of polymerisation (DP), their fraction of acetylation (FA) and their pattern of acetylation (PA).

Although the DP and FA can be resolved by MS analysis, the elucidation of the PA remains an obstacle and requires chemical labelling of the reducing end[42]. It was also shown that the PA can not be readily resolved by NMR[43]. Although the stereochemistry of this class of saccharides is well-know, we show here that the anomeric memory can be advantageously used to resolve the pattern of acetylation, without the need for purification or chemical labelling of the sample.

The MS spectrum of a crude sample of chito-oligosaccharides is shown in Fig. 3. The spectrum displays two main peaks at 789 and 992 *m/z*, which correspond to singly deacetylated tetrasaccharides and pentasaccharides, respectively. The degree of polymerisation and the fraction of acetylation are thus readily resolved. The patterns of acetylation remain unknown however: four and five candidate sequences are possible for the tetrasaccharide (dp4) and the pentasaccharide (dp5), respectively. Further MS/MS analysis of these two species shows a GlcN fragment at 180 *m/z* (marked with stars in Fig. 3), which indicates that both oligosaccharides are deacetylated at one end. On the basis of this additional information, only two candidate sequences remain. To fully resolve the sequence, the origin of the GlcN fragment (reducing or non-reducing end) must be elucidated. To that end, it is necessary to identify its anomeric configuration. Indeed, the reducing end monosaccharide would show both α and β characters (such as the GlcN standard shown in Fig. 1), whereas the non-reducing end monosaccharide would retain a pure β configuration after fragmentation of the β bond. The IR fingerprints of the GlcN fragments are displayed in Fig. 3 and compared with the relevant standards (methyl-blocked αGlcN and βGlcN) in the highly anomeric-specific spectral region 3100–3400 cm$^{-1}$. It shows clearly that the fingerprints of the GlcN fragments match this of the βGlcN standard: the β diagnostic features (highlighted in blue) are present, whereas the α diagnostic feature (highlighted in red) is absent. This indicates that both species present in the sample are deacetylated at the non-reducing end. By combining MS and spectroscopic structural information, the sequences of the oligosaccharides present in the crude sample are thus fully resolved.

## Discussion

We have demonstrated that the IRMPD signature in the 3 μm spectral range is a powerful metric to resolve carbohydrate isomerisms, and we highlight its complementarity with the structural resolution offered by ion mobility spectrometry. IRMPD allows for a further spectroscopic exploration of the structure of MS fragments. Comparison of the IR fingerprints of monosaccharide fragments with monosaccharide standards reveals two remarkable properties: the monosaccharide content can be elucidated and the stereochemistry of the glycosidic bond is retained after fragmentation.

On the basis of these findings, we propose the basic principles of a generic method for oligosaccharide sequencing. As an example of application of the method, the sequences of chito-oligosaccharides were resolved directly from a crude sample, without chemical labelling, and without the recourse to chito-oligosaccharide standards.

Compared to other MS-based sequencing strategies, the proposed method has three advantages. First, it does not require any chemical derivatisation of the analyte, thus saving considerable sample preparation. Second, a minimal database of mono-accharide standards is sufficient to perform a spectroscopic identification of the monosaccharide content and stereochemistry of any oligosaccharide. As such, this is a de novo method, in contrast with MS/MS analysis, which requires reference MS/MS data for the corresponding standard and generally involves the

availability of a large set of candidates. Finally, adding a spectroscopic dimension to mass spectrometry combines the advantages of MS-sequencing strategies with an unparalleled resolution of carbohydrate isomerisms in an integrated instrument. The method can thus be used either standalone or complement to other sequencing strategies when structural ambiguities remain. With a typical analysis time of 10 min per monosaccharide, we expect that this hybrid analytical solution will have an immediate and broad impact in glycoanalysis.

## Methods

MS is performed using a quadrupole ion trap mass spectrometer (LCQ Thermo Finnigan) coupled with electrospray ionisation source. Protonated hexosamines and $N$-acetylhexosamines are formed from 250 µM $H_2O$:MeOH (50:50) solutions. Lithium–hexoses complexes are formed from 250 µM $H_2O$:MeOH (50:50) with 50 µM LiCl solutions.

The mass spectrometer was modified for IRMPD spectroscopy to allow injection of a tuneable IR laser beam (IR OPO/OPA Laserservion pumped with a 10 Hz YAG Continuum Surelite) inside of the ion trap to perform the IRMPD spectroscopy. When the IR wavelength is resonant with a vibrational mode of the mass-selected ions, several photons are absorbed, thus increasing the internal energy of the ion, and eventually resulting in photofragmentation. Three photofragmentation mass spectra are averaged for each wavelength and the photofragmentation yield (area fragment peak/area parent peak ratio in log scale, no unit) is plotted as a function of the wavelength without further data processing. All data points shown thorough the manuscript were obtained using the same procedure. Finally, a trend line (5-points Fourier Transform rolling averaging) is added to guide the eye. The assessment of the spectral match between the fingerprint of a fragment and the library of reference IRMPD spectra of standards is done by visual inspection.

Sample preparation and computational chemistry are detailed in Supplementary Methods.

**Data availability**. All other data is available from the authors upon reasonable request. Besides, the instrument is available via the IROGLYPH platform.

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

## Acknowledgements

This work was supported by Université Lyon 1, Institut Universitaire de France and the Fédération de Recherche André Marie Ampère. This work was granted access to the HPC resources of the FLMSN, 'Fédération Lyonnaise de Modélisation et Sciences Numériques', partner of EQUIPEX EQUIP@MESO and to the 'Centre de calcul CC-IN2P3' at Villeurbanne, France. This collaborative work takes place within the Glycophysics Network (http://glyms.univ-lyon1.fr) with the support of the French Agence Nationale de la Recherche (Grant ANR-2015-MRSEI-0010)) and was supported in the UK by BBSRC: [sLoLa BB/K00199X/1] and BBSRC, EPSRC and InnovateUK: [IBCatalyst BB/M02903411 and BB/M028836/1]. SF acknowledges support from ICMG FR 2607, Labex ARCANE (ANR-11-LABX-0003-62 01) and PolyNat Carnot Institute.

## Author contributions

B.S. and G.R. did mass spectrometry and spectroscopic measurements. S.C., S.F., C.G. and S.F. did chemistry planning, chemical synthesis and biosynthesis. A.-R.A., L.B. and C.L. did ab initio simulations. I.C. supervised project. I.C. and B.S. defined analytical strategy and did article write-up.

## Additional information

**Competing interests:** The authors declare no competing financial interests.

