## [Peer Review File · Nature Communications]

REVIEWERS' COMMENTS:

Reviewer #1 (Remarks to the Author):

Compagnon et al. have improved the manuscript and reduced it considerably to focus on the main findings. It is now easy to read and understand and meets the requirements for a publication in Nature Communications. I only have two minor comments which the authors may want to address prior to publication:

1. Recently, two publications (Mucha et al., DOI: 10.1002/anie.201702896 / Masselis et al., DOI: 10.1007/s13361-017-1728-6) reported proof-of-principle experiments on fingerprinting glycans by using cryogenic IR spectroscopy, which should be mentioned briefly in the manuscript. Both experiments show that all carbohydrate isomerisms can be resolved simultaneously by investigating intact precursor ions by IR spectroscopy at low temperatures. The present manuscript is complementary to that since it focuses on fragmentation and is experimentally much less elaborate. As such, the story presented here is not weakened by the above papers. It would actually be a good chance to briefly compare both methods and to highlight advantages and drawbacks.

2. Smoothing functions: the authors tried to improve the data representation according to my suggestions, but I personally still find it difficult to actually spot real data. Especially in Figure 3 the spectra got even smaller. The individual datapoints are very small while the "trendline" is very prominent. It is not unlikely that most of the readers will mistake the trend line for real data. I would therefore encourage the authors again to increase the spot size of the raw data points. This does not change the interpretation and is only to show what is actually measured in the experiment.

Reviewer #2 (Remarks to the Author):

This revised manuscript addresses many of the concerns of all the reviewers and simplifies the presentation. The manuscript has now focused on the identification of monosaccharide content and stereochemistry, which are the two most important and novel aspects of the study. The reviewer finds the manuscript acceptable in its present form.

Please find attached the revisions made in the manuscript following the reviewer's suggestions.

Reviewer #1 (Remarks to the Author):

Compagnon et al. have improved the manuscript and reduced it considerably to focus on the main findings. It is now easy to read and understand and meets the requirements for a publication in Nature Communications. I only have two minor comments which the authors may want to address prior to publication:

1. Recently, two publications (Mucha et al., DOI: 10.1002/anie.201702896 / Masselis et al., DOI: 10.1007/s13361-017-1728-6) reported proof-of-principle experiments on fingerprinting glycans by using cryogenic IR spectroscopy, which should be mentioned briefly in the manuscript. Both experiments show that all carbohydrate isomerisms can be resolved simultaneously by investigating intact precursor ions by IR spectroscopy at low temperatures. The present manuscript is complementary to that since it focuses on fragmentation and is experimentally much less elaborate. As such, the story presented here is not weakened by the above papers. It would actually be a good chance to briefly compare both methods and to highlight advantages and drawbacks.

>>> These two references were added in the subsection «Resolution of carbohydrates isomerisms using IRMPD fingerprint» (ref 38 and 39) and the performance of these cryogenic schemes was commented.

2. Smoothing functions: the authors tried to improve the data representation according to my suggestions, but I personally still find it difficult to actually spot real data. Especially in Figure 3 the spectra got even smaller. The individual datapoints are very small while the "trendline" is very prominent. It is not unlikely that most of the readers will mistake the trend line for real data. I would therefore encourage the authors again to increase the spot size of the raw data points. This does not change the interpretation and is only to show what is actually measured in the experiment.

>>> previous fig. 3 (now Fig. 2) was edited accordingly. The datapoints were upsized.

Reviewer #2 (Remarks to the Author):

This revised manuscript addresses many of the concerns of all the reviewers and simplifies the presentation. The manuscript has now focused on the identification of monosaccharide content and stereochemistry, which are the two most important and novel aspects of the study. The reviewer finds the manuscript acceptable in its present form.

>>> no changes were made